# Pooled incidence and case-fatality of acute stroke in Mainland China, Hong Kong, and Macao: A systematic review and meta-analysis

Fan He[1]*, Irene Blackberry[1], Liqing Yao[2], Haiyan Xie[3], Tshepo Rasekaba[1], George Mnatzaganian[4,5]

1 John Richards Centre for Rural Ageing Research, La Trobe Rural Health School, La Trobe University, Albury-Wodonga, Victoria, Australia, 2 Department of Rehabilitation, The Second Affiliated Hospital of Kunming Medical University, Kunming, Yunnan, China, 3 Department of Healthcare, Peking Union Medical College Hospital, Chinese Academy of Medical Sciences and Peking Union Medical College, Beijing, China, 4 Rural Department of Community Health, La Trobe Rural Health School, La Trobe University, Bendigo, Victoria, Australia, 5 The Peter Doherty Institute for Infection and Immunity, Melbourne, Victoria, Australia

* a.f.he@outlook.com

**Data Availability Statement:** All relevant data are within the paper and its Supporting Information files.

## Abstract

### Background

Stroke incidence and case-fatality in Mainland China, Hong Kong, and Macao vary by geographic region and rates often differ across and within regions. This systematic review and meta-analysis (SR) estimated the pooled incidence and short-term case-fatality of acute first ever stroke in mainland China, Hong Kong, and Macao.

### Methods

Longitudinal studies published in English or Chinese after 1990 were searched in PubMed/Medline, EMBASE, CINAHL, Web of Science, SinoMed and CQVIP. The incidence was expressed as Poisson means estimated as the number of events divided by time at risk. Random effect models calculated the pooled incidence and pooled case-fatality. Chi-squared trend tests evaluated change in the estimates over time. When possible, age standardised rates were calculated. Percent of variation across studies that was due to heterogeneity rather than chance was tested using the $I^2$ statistic. The effect of covariates on heterogeneity was investigated using meta-regressions. Publication bias was tested using funnel plots and Egger's tests.

### Results

Overall, 72 studies were included. The pooled incidences of total stroke (TS), ischaemic stroke (IS) and haemorrhagic stroke (HS) were 468.9 (95% confidence interval (CI): 163.33–1346.11), 366.79 (95% CI: 129.66–1037.64) and 106.67 (95% CI: 55.96–203.33) per 100,000 person-years, respectively, varied according to the four economic regions (East Coast, Central China, Northeast and Western China) with the lowest rates detected in the East Coast. Increased trends over time in the incidence of TS and IS were observed (p<0.001 in both). One-month and three-to-twelve-month case-fatalities were 0.11 (95% CI:

**Funding:** GM received 2020 China Studies Seed-funding Research Grant Scheme from La Trobe University (https://www.latrobe.edu.au/) for this study. The funders had no role in study design, data collection and analysis, decision to publish, or preparation of the manuscript.

**Competing interests:** The authors have declared that no competing interests exist.

0.04–0.18) and 0.15 (95% CI: 0.12–0.17), respectively for IS; and 0.36 (95% CI: 0.26–0.45) and 0.25 (95% CI: 0.18–0.32), respectively for HS. One-month case-fatality of IS and HS decreased over time for both (p<0.001). Three-to-twelve-month fatalities following IS increased over time (p<0.001). Publication bias was not found.

## Conclusions

Regional differences in stroke incidence were observed with the highest rates detected in less developed regions. Although 1-month fatality following IS is decreasing, the increased trends in 3-12-month fatality may suggest an inappropriate long-term management following index hospital discharge.

## Registration

Registration-URL: https://www.crd.york.ac.uk/prospero/; Reference code: CRD42020170724

## Introduction

Globally, stroke is the second leading cause of death and disability after ischaemic heart disease, accounting for 11.8% of all deaths and is the third most common cause of disability worldwide [1]. Divergent trends in stroke incidence in high- and low-to-middle-income countries have been reported, showing declining and increasing incidences, respectively [2]. Predominantly cross-sectional survey data from China, the largest middle-income country with a rapidly ageing population, suggest a continuous increase in stroke incidence during recent decades [3], with first-ever stroke notifications doubling from 2002 to 2013 [4]. Within China, there are variations in stroke incidence according to a north to south divide, with highest rates in the Northeast and lowest rates in the Southeast regions [5]. Within those regions, there is considerable variation in reporting rates [6]. The pooled estimate of stroke incidence in China is unknown.

The worldwide one-month and up-to-one-year case fatality of ischaemic and haemorrhagic stroke and their temporal trends are divergent [7–9]. Chinese data show a steady decrease in in-hospital mortality following stroke [10], which may reflect improved immediate short-term stroke management over time due to advances in stroke treatment including thrombolysis, and patient care in acute stroke units. Trends in long term (up-to-one year) case fatality by different types of stroke in China are currently unknown. Furthermore, published stroke case-fatalities are inconsistent between and within regions [6], possibly resulting from differences in diagnostic, analytic or design methods, or selected samples that are not representative of the general Chinese population.

China has four economic regions: East Coast, Central China, Northeast and Western China. These regions were formed due to the rapid urbanisation, economic development, and change in Chinese regional population distributions over the past few decades [11]. Of these, the East Coast is the most developed while the Western region is the least developed. Incidence and case fatality of stroke by these regions are unknown. The aim of this novel systematic review and meta-analysis (SR) was to estimate the overall and regional pooled incidence of stroke and one-month and up-to-one-year case-fatality in mainland China, Hong Kong and Macao, using data from longtiduial studies and following strict inclusion and exclusion criteria. Pooled trends of different types of stroke over time were also estimated.

## Methods

### Search strategy and selection criteria

This SR used the World Health Organization definition of stroke ("*rapidly developed clinical signs of focal (or global) disturbance of cerebral function, lasting more than 24 hours or leading to death, with no apparent cause other than of vascular origin*") [12]. In addition, the International Statistical Classification of Diseases and Related Health Problems codes I60-I64 (10th version) or equivalent codes in the earlier versions were utilised to identify stroke. Prospective or retrospective cohort studies on adult (aged ≥18 years) populations residing in Mainland China, Hong Kong or Macao, reported in Chinese or English and published after 1990 were eligible for inclusion. When multiple studies used the same study sample, publications with the most complete data were included. Only studies with a sample of more than 100 participants were considered; studies with fewer participants were considered as case series [13]. Studies on recurrent or prevalent stroke were excluded. Studies were also excluded when denominators for estimating the incidence were not extractable.

PubMed/Medline, EMBASE, CINAHL, and Web of Science were searched for English studies while SinoMed and CQVIP were searched for Chinese studies, published from January 1, 1990 to March 13, 2021. The title and abstract were searched for all types of acute stroke and mortality whereas, the "region", and "incidence" were searched in the full text using keywords, subject headings or MeSH terms. English and Chinese peer-reviewed articles were searched using the same strategy (S1 Table). Identified systematic reviews were further hand-searched for other potentially eligible publications. Grey literature, government reports, textbook publications, and unpublished studies were not considered. Overall, 22% of study selection and data extraction was randomly and independently reassessed. Inconsistencies in study selection were discussed in team meetings to reach consensus. If required, authors of identified papers were contacted for additional information before decisions on eligibility were reached.

### Registration and protocol

The SR was registered in International Prospective Register of Systematic Reviews [registration code: CRD42020170724]. Details of the study have been reported in the study protocol [14].

### Statistical analysis

Study variables included publication details, publication language, study design, type of stroke, diagnostic criteria, follow-up time, study period, regions, setting, sample size, stroke events, stroke severity (National Institutes of Health Stroke Scale and Glasgow Coma Scale), deaths, age, sex, smoking status and inclusion and exclusion criteria of the original study. Incidences of first ever total stroke, ischaemic stroke and haemorrhagic stroke were estimated as events per 100,000 person-years. Using the Exact method, the incidence rates of disease were expressed as Poisson means divided by time at risk. The log incidence rates together with their corresponding log standard errors, stratified by the four economic regions, were meta-analysed using DerSimonian and Laird random effects model. Random effects models were also used to estimate the pooled case fatality estimates. Region-specific analyses of pooled estimates of linear trends over time were assessed using a chi-squared test. The $I^2$ statistic assessed the heterogeneity of the included studies. Since some studies excluded patients with major comorbidities, subgroup analyses were conducted according to papers that had and did not have this exclusion criterion. Funnel plots and Egger's tests examined publication bias. Separate analyses were conducted for ischaemic, haemorrhagic, and total stroke. Meta-regressions were constructed to investigate and quantify the proportion of between-study variance explained by

known study variables. The meta regression models accounted for age, sex, smoking status, severity of stroke, study setting (community or hospital), follow-up time and risk of bias.

The Newcastle-Ottawa Scale was used to assess the quality and risk of bias. The Healthcare Research and Quality standards were used to further categorise the results of Newcastle-Ottawa Scale assessments into good, fair, or poor quality. Sensitivity analyses were conducted by risk of bias.

Stata/SE 15 was used to conduct the analyses. Stata's "metan" and "metaprop"commands were used to run the meta-analysis.

## Results

A total of 41,294 English, and 46,339 Chinese publications were identified, of which, 32,355 duplicates were removed, leaving 55,278 papers for title and abstract screening. Of these, the full text of 2,440 English and 2,720 Chinese articles was examined, with 72 papers reporting incidence and/or mortality included in the final review with one study reporting both. The reporting of this SR complied with the Preferred Reporting Items for Systematic Reviews and Meta-Analyses (PRISMA) (Fig 1).

Characteristics of the included publications (earliest published in 1997 and latest in 2020) are displayed in Table 1A and 1B. Studies reported on the incidences of total stroke (TS) (n = 28), haemorrhagic stroke (HS) (n = 20) and ischaemic stroke (IS) (n = 20). All except one study followed a prospective cohort design with the exception being retrospective cohort. Of the four economic regions represented in the incidence studies, the east coast was mostly represented (14 studies), followed by western China (five studies), northeast China (four studies), and central China (two studies) and nine were national. In total, the studies reporting incidence included 1,809,252 individuals contributing 13,060,309 person-years of follow-up. Both sexes were represented in most studies with just two studies focusing exclusively on either males or females. Sex was not identified in five incidence studies. The publication years of the 39 studies reporting on case-fatality [IS (n = 28 studies), HS (n = 14 studies)] ranged from 2003 to 2020. Case-fatalities of stroke patients in the East Coast were mostly reported in sixteen studies, followed by Central China (nine studies), Western China (five studies), and Northeast China (four studies). Five studies were national. Sex was not identified in eight case-fatality studies.

Of the 34 studies reporting incidence, 22 (65%) were classified as having good to moderate quality while 12 (35%) were deemed poor quality. The 39 papers reporting case-fatality included 31 (79%) with good to fair quality while 8 (21%) were rated as poor. The quality assessment details can be found in S2 Table. The percentage of agreement among co-authors (FH, HX, TR, GM) and collaborators (YZ, YH, ZP, SL, RY) conducting the culling and quality assessments ranged from 88% to 95%, with the corresponding Cohen Kappa values ranging from 0.48 to 0.56. The overall percentage of agreement was 92%, Cohen Kappa 0.51 [95% confidence interval (CI): 0.45–0.58].

Geographical differences in the pooled incidences of total stroke were found with the highest being in Central China (887.87 per 100,000 person-years) and the lowest in the East Coast region (409.34 per 100,000 person-years). The overall incidence was 468.90 (95% CI: 163.33–1346.11) per 100,000 person-years (Fig 2), increasing to 541.01(95% CI: 149.51–1957.68) per 100, 000 person-years when poor quality studies were excluded from the analysis. The combined pooled estimate in Central, Northeast and Western China of 698.52 (95% CI: 74.87–6517.18) per 100,000 person-years was significantly higher than that in the East Coast region (409.34 (95% CI: 73.22–2288.38)), p<0.001 (S1 Fig).

Similar regional differences were detected for both ischaemic and haemorrhagic stroke when comparing overall pooled estimates for Central, Northeast and Western China with the

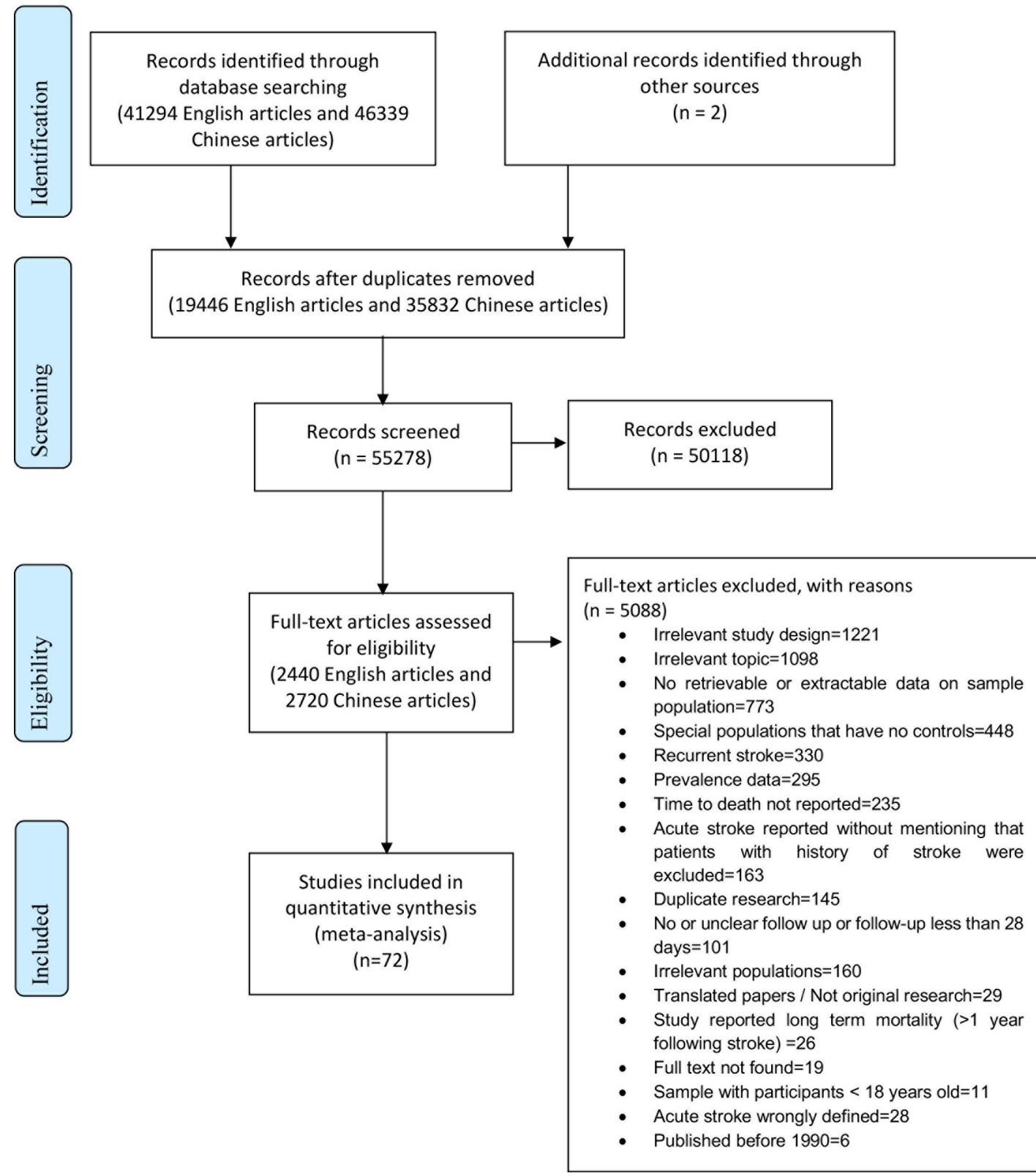

**Fig 1. PRISMA chart, study identification.**

**Table 1. a. Description of included studies: incidence.** b. Description of included studies: case-fatality.

a

| Study | Publication language | Economic region† | Period of study | Risk of bias | Mean age | % Male | Sample size | Type of stroke and number of cases | Follow-up (person-years) | Mean follow-up (years) |
|---|---|---|---|---|---|---|---|---|---|---|
| Du et al (1997) [15] | Chinese | East Coast | 1979–1993 | Fair | 48.6 | 50.6 | 1809 | 35 TS, 16 HS | 25212 | NR |
| Duan et al (2019) [16] | English | East Coast | 2010–2016 | Fair | 49.5 | 100 | 23433 | 678 TS, 595 IS, 109 HS | 138255 | 5.9 |
| Gu et al (2020) [17] | English | East Coast | 2009–2014 | Poor | 69.1 | 43.8 | 1257 | 113 TS | 6136 | 4.8 |
| Guo et al (2013) [18] | Chinese | East Coast | 2003–2010 | Good | 56.2 | 42.1 | 10565 | 195 TS | 67886 | 6.4 |
| Jia et al (2011) [19] | English | East Coast | 2006 –NR | Poor | 52.9 | 52.8 | 19369 | 341 TS | 58107 | 3 |
| Li et al (2015) [20] | English | East Coast | 1992–2012 | Poor | NR | NR | 3031 | 908 TS, 664 IS, 231 HS | 304260 | NR |
| Liu et al (2014) [21] | English | East Coast | 2006–2010 | Fair | 51.4 | 79.5 | 90517 | 1472 TS, 1049 IS, 423 HS | 362163 | 4.1 |
| Wang et al (2016) [22] | English | East Coast | 2002–2012 | Good | 56 | 34.8 | 1263 | 116 TS, 101 IS | 11569 | 9.2 |
| Wang et al (2018) [23] | English | East Coast | 2009–2016 | Fair | 60.2 | 40.6 | 4081 | 138 IS | 20040 | 5.2 |
| Wang et al (2019) [24] | Chinese | East Coast | 2005–2013 | Fair | NR | 36.4 | 29689 | 1716 IS, 91 HS | 175667 | 6.7 |
| Yu et al (2016) [25] | English | East Coast | 1996–2011 | Fair | 51.5 | 0 | 64328 | 2991 TS, 2750 IS, 241 HS | 618412 | 10 |
| Zhang et al (2019) [26] | English | East Coast | 2010–2015 | Fair | 58.4 | 37.8 | 8500 | 444 TS | 43350 | 5.1 |
| Ji et al (2020) [27] | English | East Coast | 2009–2017 | Poor | 45.7 | 47.8 | 27712 | 617 TS | 193984 | 7 |
| Wang et al (2020) [28] | Chinese | East Coast | 2005–2013 | Poor | 52.1 | 36 | 28940 | 2164 IS, 127 HS | 177279 | 6.2 |
| Han et al (2017) [29] | English | Central | 2001–2012 | Fair | 71.7 | 48.2 | 2852 | 211 TS | 18628 | NR |
| Zhou et al (2020) [30] | English | Central | 2008–2014 | Fair | 61.7 | 44.1 | 31750 | 1557 TS, 1151 IS, 287 HS | 196850 | 6.2 |
| Li et al (2018) [31] | English | Northeast | 2008 –NR | Poor | NR | NR | 1928 | 98 TS | 9531 | 4.9 |
| Sun et al (2013) [32] | English | Northeast | 2004–2010 | Fair | NR | NR | 38949 | 858 TS, 483 IS, 348 HS | 168023 | NR |
| Xie et al (2019) [33] | English | Northeast | 2012–2017 | Fair | 55.6 | 34.6 | 3229 | 81 TS | 15499 | 4.8 |
| Yu et al (2020) [34] | English | Northeast | 2004–2014 | Fair | 50.5 | NR | 38589 | 3855 TS | 275526 | 7.1 |
| Guo et al (2016) [35]* | English | Western | 2002–2012 | Poor | 54 | 52.4 | 425901 | 13274 IS, 2917 HS | 1895447 | NR |
| Olofindayo et al (2015) [36] | English | Western | 2003–2012 | Fair | 46.5 | 40.9 | 2589 | 121 TS, 75 IS, 44 HS | 23292 | 9.2 |
| Ren et al (2019) [37] | Chinese | Western | 2011–2015 | Poor | 46.8 | 61 | 33042 | 448 TS | 80506 | 2.4 |
| Zhang et al (2009) [38] | English | Western | 2002–2008 | Fair | 64.3 | 45.2 | 2173 | 82 TS, 52 IS, 30 HS | 9952 | 4.6 |
| Zhou et al (2016) [39] | Chinese | Western | 2011–2014 | Poor | NR | NR | 4160 | 51 TS | 13621 | 3.3 |
| Huang et al (2003) [40] | Chinese | National | 1994–2001 | Poor | 54.7 | 51.8 | 24475 | 228 TS, 151 IS, 77 HS | 105201 | 4.3 |
| Huang et al (2019) [41] | English | National | 1992–2015 | Good | 50.9 | 41 | 117575 | 2276 IS, 1019 HS | 900214 | NR |
| Kelly et al (2008) [42] | English | National | 1991–2000 | Good | 55.9 | 49.1 | 155131 | 2421 HS | 1287587 | 8.3 |
| Liu et al (2008) [43] | Chinese | National | 1992–2003 | Poor | 46.9 | 53.5 | 30384 | 613 TS, 427 IS, 159 HS | 199711 | NR |
| Wang et al (2013) [44] | English | National | 1987–1998 | Fair | 51.6 | 47.2 | 26607 | 1108 TS, 614 IS, 451 HS | 241149 | NR |
| Zhang et al (2007) [45] | Chinese | National | 1992 –NR | Poor | NR | 46.4 | 15131 | 370 TS, 266 IS, 107 HS | 163415 | 10.8 |

*(Continued)*

**Table 1.** (Continued)

| Zhou et al (2003) [46] | Chinese | National | 1991–1997 | Fair | 45 | 48.6 | 9111 | 259 TS, 177 IS, 84 HS | 131188 | NR |
| Chen et al (2020) [47] | English | National | 2004–2017 | Fair | 51.6 | 40.9 | 489586 | 45732 TS, 36588 IS, 8142 HS | 4720000 | NR |
| Swaminathan et al (2020) [48] | English | National | 2003–2019 | Fair | 50.5 | 41.9 | 41596 | 1752 TS | 402649 | 9.7 |

b

| Study | Publication language | Economic region† | Type of stroke | Period of study | Risk of bias | Mean age | % Male | Cases | Deaths | Follow-up |
|---|---|---|---|---|---|---|---|---|---|---|
| Liu et al (2018) [49] | Chinese | East Coast | IS | 2013–2017 | poor | 68 | 54.5 | 101 | 14 | 3 months |
| Shi et al (2019) [50] | English | East Coast | IS | 2010–2017 | good | NR | NR | 334 | 28 | 3 months |
| Geng et al (2016) [51] | Chinese | East Coast | IS | 2013–2015 | fair | NR | NR | 796 | 37 | 3 months |
| Fang et al (2012) [52] | English | East Coast | IS | 2003–2008 | good | 66.7 | 56.3 | 1184 | 113 | 28 days |
| Tu et al (2017) [53] | English | East Coast | IS | 2015 | fair | 58 | 54.3 | 737 | 104 | 3 months |
| Zi et al (2013) [54] | English | East Coast | IS | 2009–2012 | fair | 65 | 65.9 | 226 | 34 | 3 months |
| Chen et al (2016) [55]* | English | East Coast | IS | 2013–2015 | good | 68.4 | 58.4 | 871 | 94 | 3 months |
| Huang et al (2013) [56] | English | East Coast | IS | 2008–2011 | fair | 66.3 | 58 | 338 | 68 | 12 months |
| Liu et al (2020) [57]* | Chinese | East Coast | IS | 2012–2019 | poor | 72 | 50.4 | 117 | 20 | 3 months |
| Xi et al (2020) [58] | Chinese | East Coast | IS | 2016–2019 | poor | 73 | 54.3 | 138 | 16 | 3 months |
| Hong et al (2016) [59] | Chinese | East Coast | IS | not given | fair | NR | NR | 2479 | 255 | 3 months |
| | | | | | | | | 2479 | 349 | 12 months |
| Wang et al (2018) [60]* | English | East Coast | HS | 2016–2017 | fair | 65.8 | 61.9 | 181 | 35 | 1 month |
| Wei et al (2014) [61] | English | East Coast | HS | 2010–2012 | fair | 69 | 53.1 | 271 | 34 | 3 months |
| Yan et al (2016) [62] | English | East Coast | HS | 2011–2014 | poor | 63.2 | 58.9 | 112 | 33 | 6 months |
| Jiang et al (2018) [63] | Chinese | East Coast | HS | 2016–2017 | fair | 63 | 66.3 | 172 | 12 | 30 days |
| Hu et al (2012) [64] | English | East Coast | HS | 2011 | fair | 63.1 | 61.4 | 176 | 64 | 3 months |
| Nie et al (2017) [65] | English | Central | IS | 2013–2016 | fair | 59 | 53 | 387 | 74 | 12 months |
| Wang et al (2014) [66] | English | Central | IS | 2011–2014 | fair | 68 | 50.9 | 275 | 75 | 12 months |
| Zhu et al (2018) [67] | English | Central | IS | 2016–2017 | fair | 66 | 48.2 | 299 | 31 | 3 months |
| Jing et al (2019) [68] | English | Central | IS | 2016–2018 | fair | 65 | 58.2 | 196 | 43 | 12 months |
| Sun et al (2015) [69] | English | Central | IS | 2004–2010 | good | NR | NR | 790 | 73 | 30 days |
| | | | | | | | | 790 | 116 | 12 months |
| Lin et al (2014) [70] | Chinese | Central | HS | 2010–2012 | fair | 58.2 | 62.9 | 167 | 58 | 6 months |
| Zhang et al (2014) [71]* | English | Central | HS | 2007–2013 | poor | 72.6 | 59.3 | 118 | 50 | 1 month |
| Yang et al (2004) [72] | English | Central | HS | not given | poor | 69.7 | NR | 722 | 407 | 28 days |
| Jiang et al (2011) [73] | Chinese | Central | HS | 2009–2010 | fair | 61.5 | 55.5 | 182 | 57 | 30 days |
| Wang et al (2014) [74] | English | Northeast | IS | 2012–2013 | fair | 65 | 60.4 | 326 | 38 | 1 month |
| Dong et al (2013) [75] | English | Northeast | IS | 2010–2011 | poor | 69 | 55.2 | 125 | 18 | 3 months |
| Zhang et al (2013) [76] | English | Northeast | IS | 2007–2011 | fair | 72 | 58 | 245 | 41 | 12 months |
| Xie et al (2016) [77] | English | Northeast | IS | 2013–2014 | fair | 61 | 64.8 | 216 | 32 | 6 months |
| Du et al (2018) [78] | Chinese | Western | IS | 2013–2017 | poor | 70.8 | 55.7 | 122 | 34 | 3 months |

(Continued)

**Table 1.** (Continued)

| | | | | | | | | | | |
|---|---|---|---|---|---|---|---|---|---|---|
| Zhang et al (2007) [79]* | English | Western | IS | 2000–2006 | fair | NR | NR | 1904 | 327 | 1 month |
| Tao et al (2017) [80] | English | Western | HS | 2010–2013 | fair | 58.5 | 64.3 | 336 | 90 | 3 months |
| He et al (2018) [81] | English | Western | HS | 2015–2016 | fair | NR | NR | 221 | 47 | 3 months |
| Li et al (2008) [82] | English | Western | IS | not given | fair | 65.5 | 57 | 395 | 32 | 3 months |
| | | | HS | not given | | 65.5 | 57 | 254 | 39 | 3 months |
| Wu et al (2014) [83] | English | Nation wide | IS | 2007–2010 | fair | 63 | 64.9 | 1229 | 144 | 12 months |
| Tu et al (2017) [84] | English | Nation wide | IS | 2012–2015 | fair | 68 | 54.4 | 4215 | 906 | 12 months |
| Tu et al (2019) [85] | English | Nation wide | IS | 2015–2017 | fair | 65 | 52 | 1530 | 325 | 6 months |
| Zhang et al (2003) [86] | English | Nation wide | IS | 1991–2000 | good | NR | NR | 6772 | 1144 | 28 days |
| | | | HS | 1991–2000 | | NR | NR | 3489 | 1691 | 28 days |
| Chen et al (2020) [47] | English | Nation wide | IS | 2004–2017 | fair | 59.3 | 44.8 | 36588 | 1098 | 28 days |
| | | | | | | 59.3 | 44.8 | 36588 | 2267 | 12 months |
| | | | HS | 2004–2017 | fair | 59.3 | 51.3 | 8142 | 3630 | 28 days |

Notes

*Studies marked with asterisk followed a retrospective design, while others were prospective cohort.

†Four economic regions: East Coast (Beijing, Tianjin, Hebei, Shanghai, Jiangsu, Zhejiang, Fujian, Shandong, Guangdong and Hainan); Central (Shanxi, Anhui, Jiangxi, Henan, Hubei and Hunan); Northeast (Liaoning, Jilin and Heilongjiang); Western (Inner Mongolia, Guangxi, Chongqing, Sichuan, Guizhou, Yunnan, Tibet, Shaanxi, Gansu, Qinghai, Ningxia and Xinjiang)

Abbreviation: Not reported (NR)

East Coast region. For IS the pooled estimate of 505.26 (95% CI: 49.47–5160.46) compared to 438.33 (95% CI: 66.01–2910.54) per 100,000 person-years for the East Coast Region while, for HS, the pooled estimate of 165.08 (95% CI: 31.91–854.01) compared to 57.41 (95% CI: 18.24–180.65) per 100,000 person-years for the East Coast region (S2 and S3 Figs). No between-studies heterogeneity was detected in all types of stroke and no publication bias was found as shown in S4 Fig. Egger's tests p values for TS, IS and HS were p = 0.17, p = 0.39, and p = 0.43, respectively. The regional differences observed in the sensitivity analysis were consistent with our main findings (S5 Fig). Age standardisation was limited to a few selected studies based on data availability resulting in an estimated age standardised incidence for TS, IS, and HS of 285.76 (95% CI: 29.37–2780.33), 586.76 (95% CI: 7.05–48813.46) and 141.55 (95% CI: 4.58–4379.29) per 100, 000 person-years, respectively.

Fig 3 shows the scatter plots for stroke incidence using the median study period year for each study. Studies published in a period of four consecutive years were pooled together, starting from the latest year. The analyses showed a statistically significant increase over time in total stroke, ischaemic stroke and haemorrhagic stroke (Chi-squared test p value<0.001 in all).

Subgroup analyses were based on initial study sample selection criteria. Studies that included a more representative sample (without the exclusion of patients with major comorbidities such as coronary heart disease) were categorised as group one. Studies that used selected samples, often only including healthier individuals, which made the samples less representative of the general Chinese population, were categorised as group two. Pooled incidence of TS in the more representative sample was slightly higher than that found in group two, 474.09 (95% CI: 38.47–5842.50) per 100,000 person-years versus 467.79 (95% CI: 146.36–1495.20) (S6 Fig).

The pooled estimates of one-month and three-to-twelve-month case-fatality were 0.11 (95% CI: 0.04–0.18) and 0.15 (95% CI: 0.12–0.17), respectively for IS; and 0.36 (95% CI: 0.26–

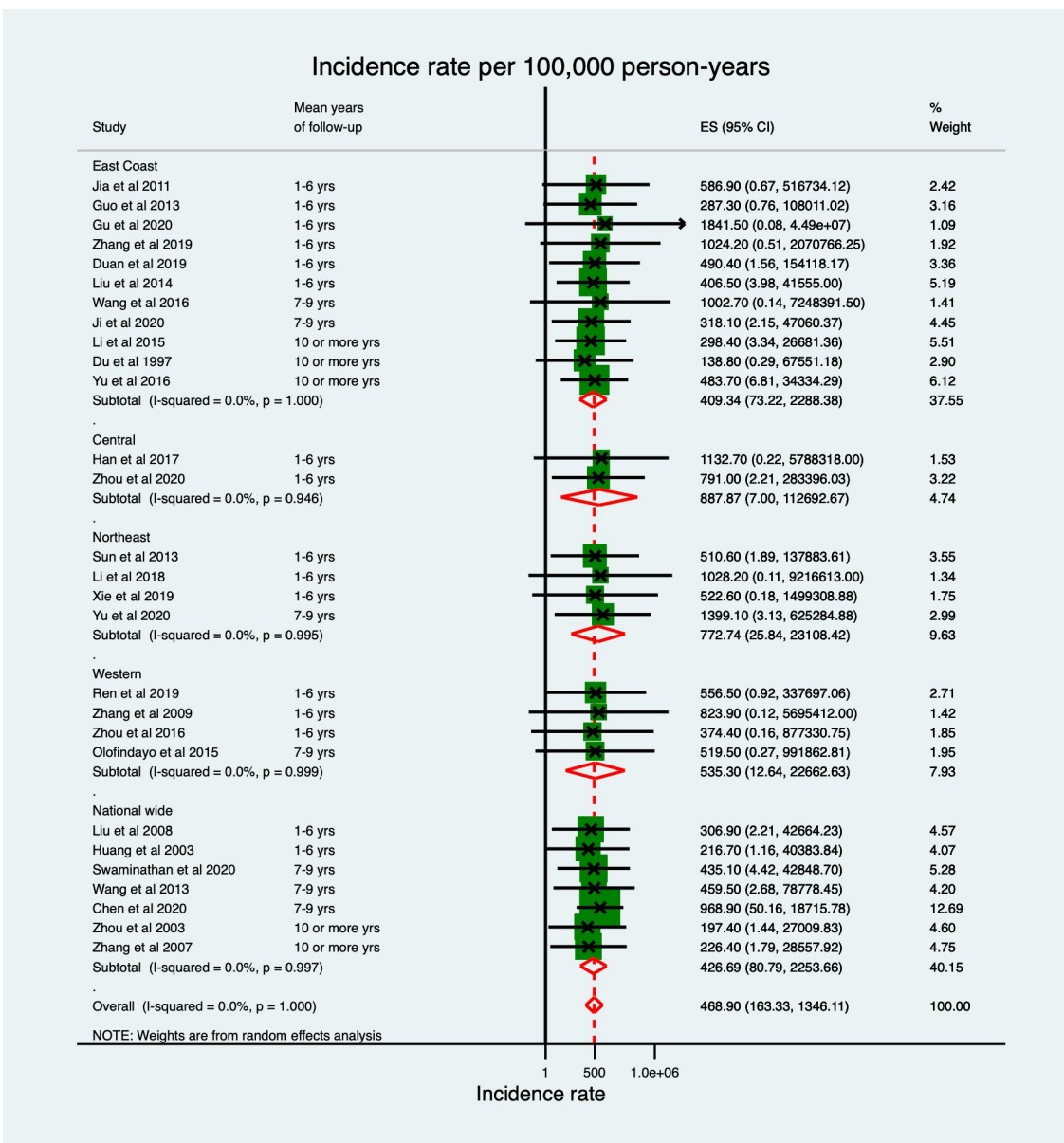

**Fig 2. Total stroke incidence rate per 100,000 person-years.**

0.45) and 0.25 (95% CI: 0.18–0.32), respectively for HS (Figs 4 and 5). Excluding studies with poor quality produced similar findings. High between-study heterogeneity was found when studies were stratified by follow-up time ($I^2$ >90%). Based on meta-regressions, risk of bias, study setting (community or hospital), and follow-up time explained between 11.4% to 17.4% of between-study variance in IS. For HS, risk of bias, study setting, and smoking status explained between 51.7% to 73.5% of between-study variance. Mean age, sex, and severity of stroke did not contribute to study heterogeneity in either type of stroke.

Following the same method used in subgroup analysis of stroke incidence, one-month estimates of IS were similar in both groups. However, these varied for HS with pooled estimates being higher in the more representative sample [0.49 (95% CI: 0.44–0.55) versus 0.25 (95% CI: 0.10–0.40)] in the less representative group.

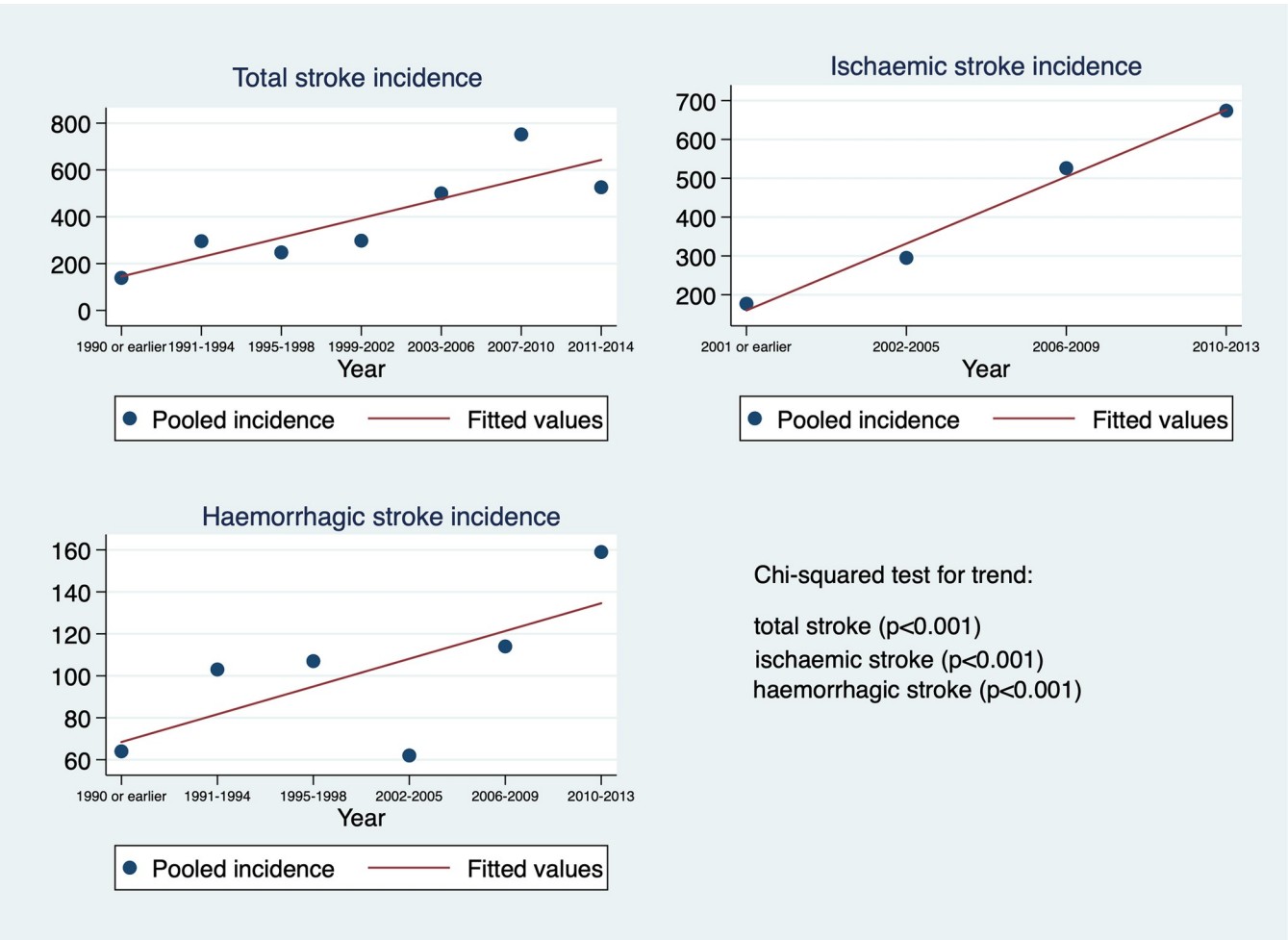

**Fig 3. Scatter plots of stroke incidence by median study period.**

The scatter plots of case-fatality of IS and HS using the median study period year for each study are shown in Fig 6. Decreased trends in 1-month IS and HS case-fatality were detected (p<0.001 in both), whereas, 3-12-month IS case-fatality increased over time (p<0.001). Trends in 3-12-month HS case fatality could not be investigated due to limited points in time (n = 3) after studies were pooled by median years.

## Discussion

This is the first study to report pooled estimates of incidence and case-fatality of stroke in China using data from longitudinal cohort studies. Our estimated figures support previous findings that China has one of the highest rates of stroke incidence in the world [87]. We report significant differences in incidences of TS, IS and HS between the East Coast (with lowest rates) and other economic regions (with higher rates) in China. This SR reports that although 1-month fatality following ischaemic stroke has decreased over the past decades, 3-12-month mortality following the event has significantly increased during this time.

Rates of stroke incidence rates are often reported inconsistently in China [6]. Such inconsistensies in reportings may have resulted from different diagnostic and definitional criteria [6], and different study designs including door-to-door surveys that lack follow-up periods and

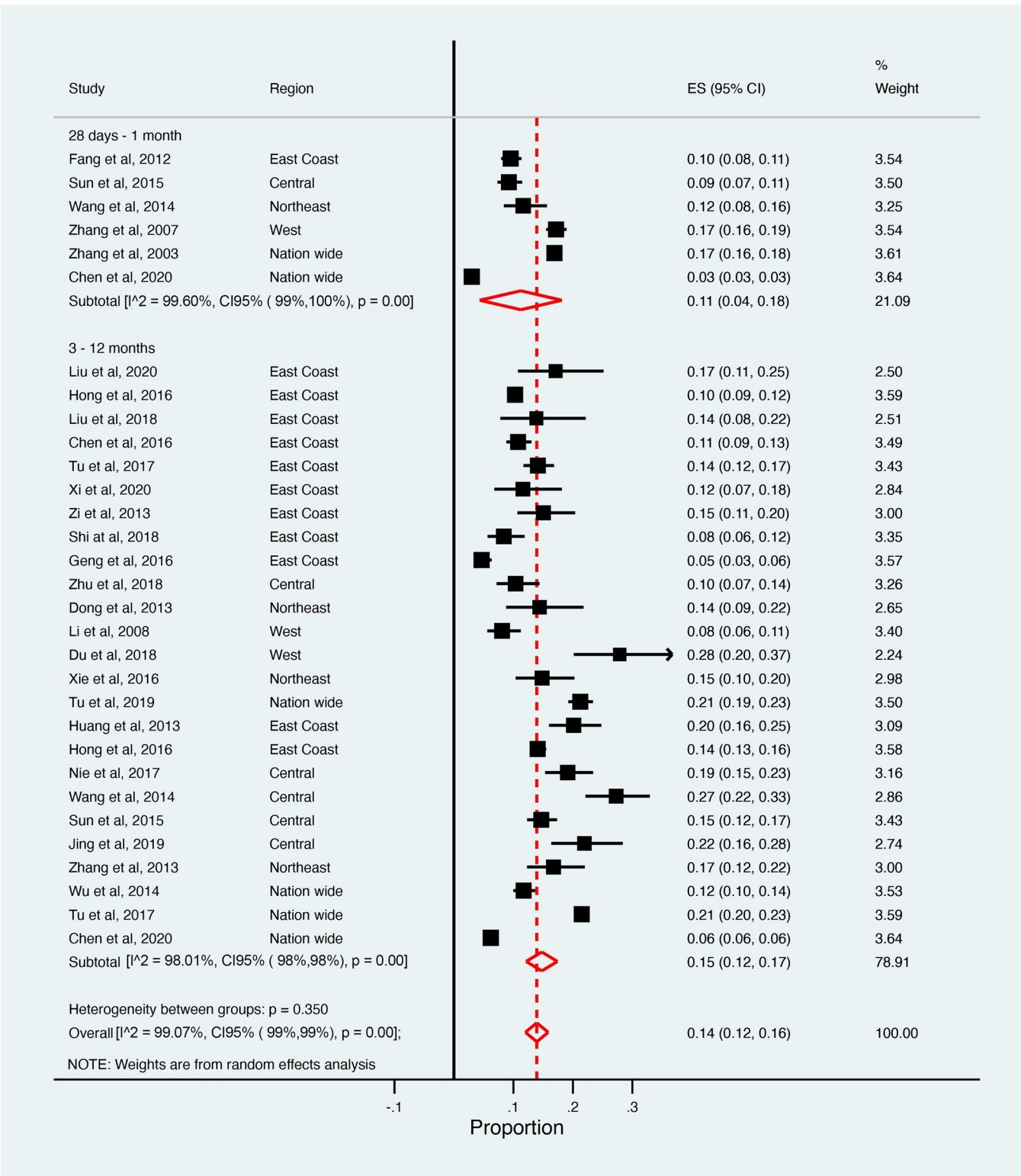

**Fig 4. Pooled case-fatalities of ischaemic stroke.**

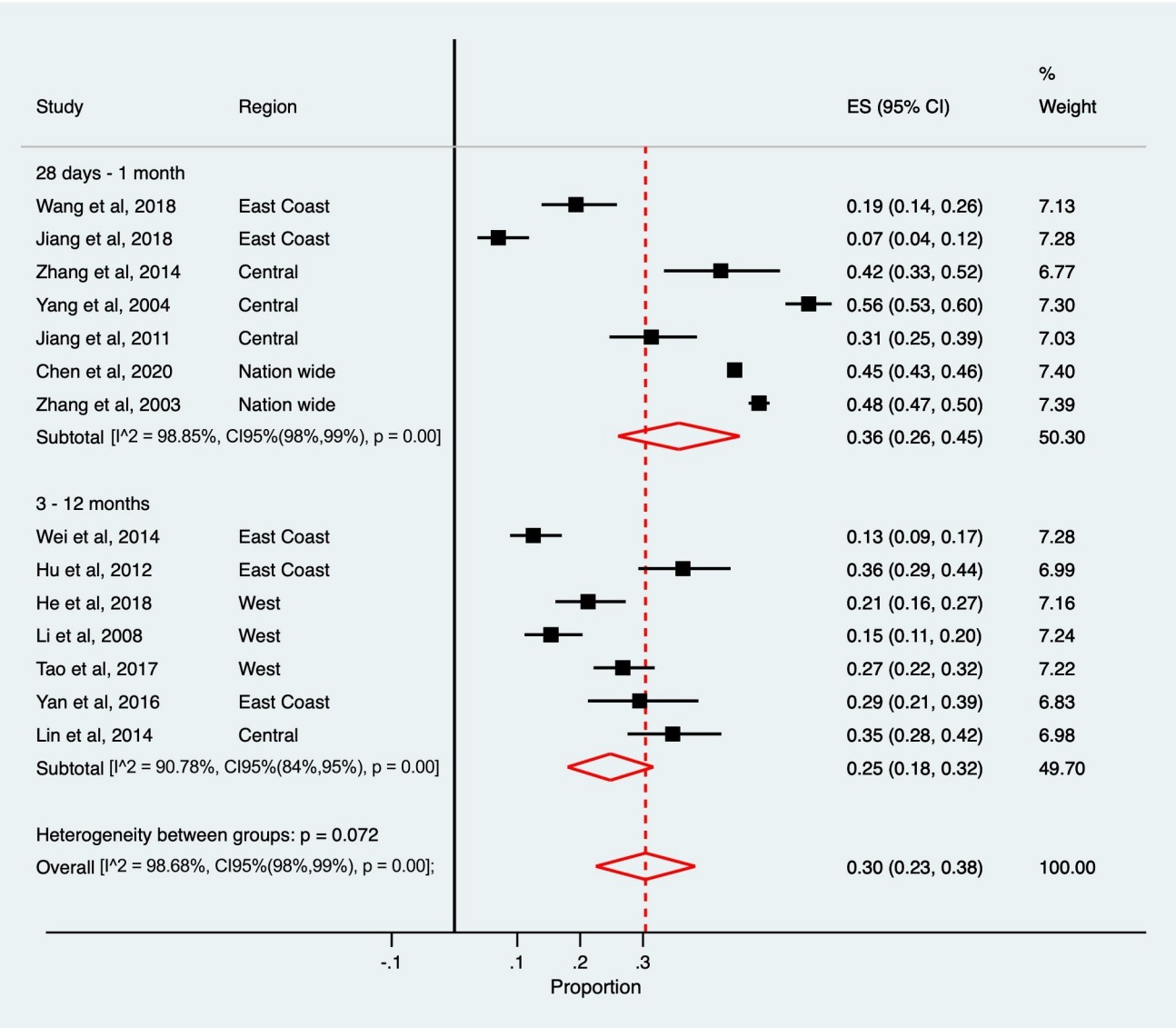

**Fig 5. Pooled case-fatalities of haemorrhagic stroke.**

potentially generate biased estimates due to sampling errors, non-response and reporting bias [88]. As detected in our SR, studies reporting stroke incidence have often excluded patients with histories of certain comorbidities [27], making the sample less representative of the general Chinese population at risk of stroke. Regional differences in stroke incidence may also be explained by the different patterns of risk factors reported across China. For instance, the higher prevalence of hypertension in Northern, Northeast and Central China may have contributed to a higher incidence of stroke [3]. The disparities in stroke incidence among the four economic regions, identified in this SR, highlight the need for primary stroke prevention, health promotion, and better access to health services in less-developed regions.

Consistent with previous research [4], this SR reports an increased incidence of TS, IS and HS over time, which is consistent with findings from other low to middle countries that are experiencing a similar trend [2]. As a result of rapid economic development, there is evidence to indicate that populations living in different parts of China have changed their lifestyle, their

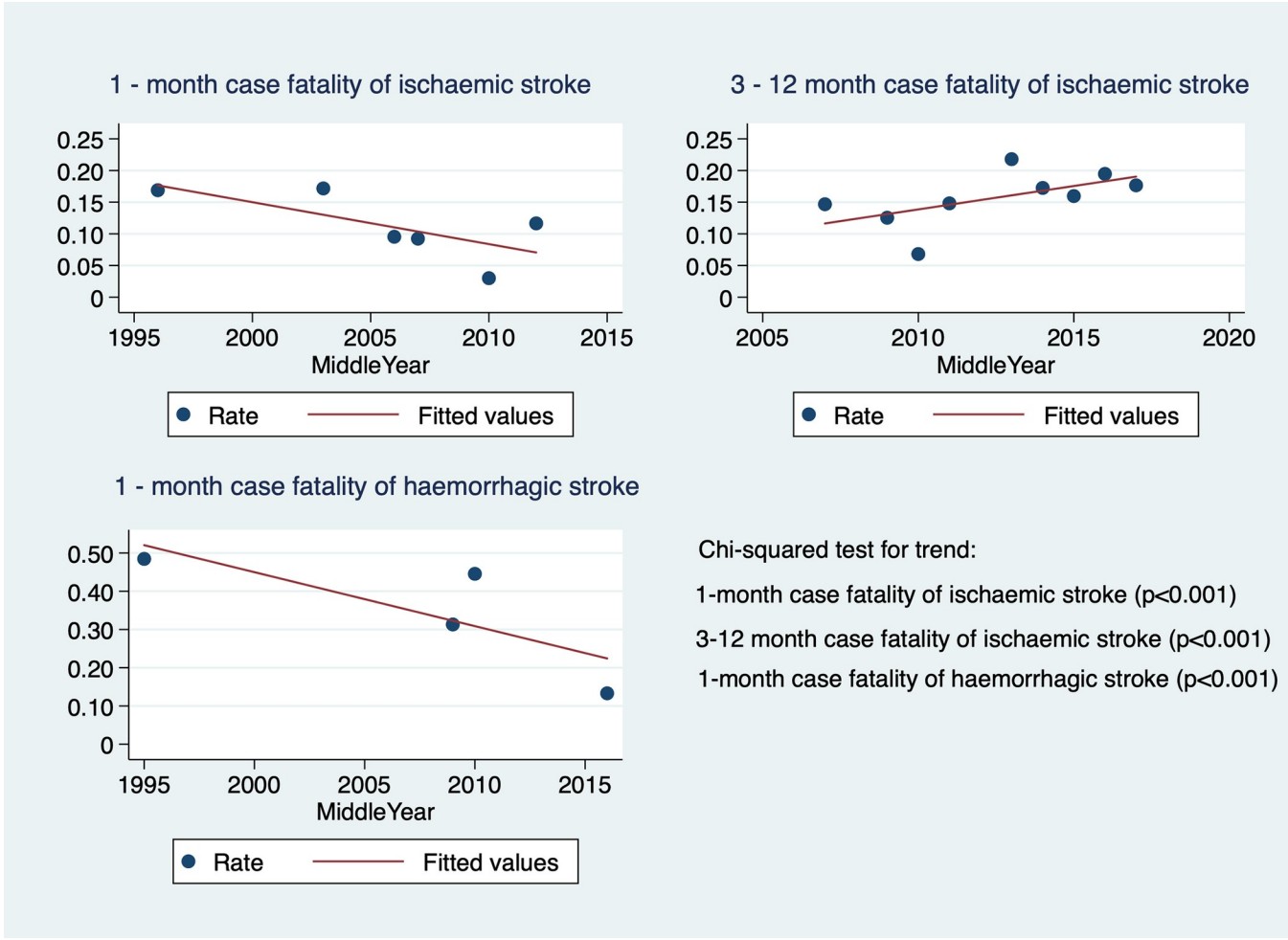

**Fig 6. Scatter plots of stroke case-fatality by median study period.**

diet (consuming more fast foods), and have become less physically active [89]. This may have led to higher rates of obesity, hypertension and diabetes, which in turn can lead to higher rates of stroke [90]. The increase in incidence may also be explained by the rapidly aging population or improvements in stroke diagnosis [4, 91]. In contrast, stroke incidence in high income countries significantly declined from 1990 to 2010 [2]. This may indicate that China lags behind in stroke prevention despite the rapid economic growth experienced in the same period.

This SR reports decreasing trends in one-month IS and HS case-fatality, consistent with a previous study that reported decreased in-hospital mortality for all types of stroke [10]. Underlying factors contributing to the downward trend in short-term stroke case-fatality in China may include improved treatments, expanded healthcare coverage, enhanced preventative campaigns and increased public health literacy around stroke [92]. This study, however, also found an opposing increase in 3-12-month IS case-fatality, which may indicate a need to improve secondary prevention, post-stroke rehabilitation and support for caregivers, especially for those discharged from acute care. This is supported by a study that found that Chinese stroke patients were less likely to adhere to antiplatelet and lipid lowering therapy after 3 months post discharge, with a large proportion of these patients not filling prescriptions or refusing to take prescribed medications [93]. Barriers for patients to adhere to post-stroke medication

regimens include limited health literacy, inadequate communication and poor coordination between medical staff and patients, and cost [94]. Although recommended by guidelines, a national study reported that around 40% of IS patients were not properly rehabilitated with rehabilitation services varying across hospitals [95]. Underlying reasons for such suboptimal stroke rehabilitation may include lack of insurance coverage, absence of well-established rehabilitation systems, a lack of trained and licensed therapists and caregivers, and limited availability of rehabilitation technologies [96]. Furthermore, due to the barriers to accessing government-funded services, post-stroke patient care often falls on family members who lack medical skills and have little social support to properly manage discharged patients at home [97, 98].

This SR highlights the need to tackle inequality in stroke prevention across the four economic regions by reinforcing and concretising policies. This SR also emphasises the need to establish well-coordinated rehabilitation system programs that will meet the needs of individuals coming from different socioeconomic backgrounds.

## Strengths and limitations

The comprehensive search strategy targeted six major electronic databases to identify potentially eligible English and Chinese articles. The earliest publication year was restricted to 1990 since brain imaging technologies were widely used after this year in China [99], which made the case ascertainment of stroke more reliable. The restriction to 1991 onwards aimed to decrease false positives and to increase the validity of included cases. Our strict selection criteria, that included longitudinal studies with a defined follow-up period, ensured the inclusion of only acute first-ever stroke. This study, covering a time period of four decades (1979 to 2019), included around 2 million participants of all ages and over 13 million person-years of follow-up.

This SR also has limitations. Less developed regions were under-represented which may indicate that the regional differences we found could, in reality, be greater. The age-standardised rates were mostly based on studies from the East Coast, which had lower estimates than other regions. This may explain the large differences in the crude and age-standardised rates. Duplicate research populations were carefully screened for using all available information with papers reporting on the same study population included once. There may, however, have been some overlap in study populations from the same region in studies which included limited descriptions of participants. Subtypes of IS and HS were not evaluated in this SR. The sources of heterogeneity investigated in our study were limited to available variables reported in the included studies.

## Conclusions

The disparities in stroke incidence between economically developed regions and less developed regions necessitate decision-making and action to address inequalities in stroke education and prevention across China. The rising trends in 3-12-month case-fatality following ischemic stroke highlight the need to develop more effective secondary stroke prevention programs, including higher adherence to antiplatelet and lipid lowering therapies and improvement of post stroke rehabilitation.

## Supporting information

**S1 Checklist. PRISMA checklist.**
(DOCX)

**S1 Table. Search strategy.**
(DOCX)

**S2 Table. Risk of bias assessments.**
(DOCX)

**S1 Fig. Pooled estimates of total stroke incidence in east coast China and other regions.**
(TIFF)

**S2 Fig. Pooled estimates of ischaemic stroke incidence in east coast China and other regions.**
(TIFF)

**S3 Fig. Pooled estimates of haemorrhagic stroke incidence in east coast China and other regions.**
(TIFF)

**S4 Fig. Funnel plots and Egger's tests.**
(TIFF)

**S5 Fig. Sensitivity analysis for total stroke incidence.**
(TIFF)

**S6 Fig. Total stroke incidence rate by sub-group analysis.**
(TIFF)

**S7 Fig. Pooled estimates of total stroke incidence in map of China.**
(TIFF)

**S8 Fig. Total stroke incidence by length of follow up.**
(TIFF)

**S9 Fig. Pooled estimates of ischaemic stroke incidence by the four Chinese economical regions.**
(TIFF)

**S10 Fig. Pooled estimates of haemorrhagic stroke incidence by the four Chinese economical regions.**
(TIFF)

**S1 File. Included Chinese papers and translations.**
(DOCX)

## Acknowledgments

This study was conceived and designed by FH, GM and IB. Search strategy was developed by FH, GM, LY and HX. Articles selection and data extraction were conducted by FH, GM, TR and HX. Data analysis were run, and results were interpreted by FH and GM. Manuscript was drafted by FH, GM, reviewed and approved by all authors.

We thank Ms Ying Zhao, Ms Ying He, Mr Zhipeng Pan, Mr Sicheng Li and Ms Ruihong Yao for their contribution in study selection. We thank Dr Sean MacDermott for proofreading the final version of the manuscript.

## Author Contributions

**Conceptualization:** Fan He, Irene Blackberry, George Mnatzaganian.

**Data curation:** Fan He.

**Formal analysis:** Fan He, George Mnatzaganian.

**Funding acquisition:** Irene Blackberry, George Mnatzaganian.

**Investigation:** Fan He, Haiyan Xie, Tshepo Rasekaba, George Mnatzaganian.

**Methodology:** Fan He, Irene Blackberry, Liqing Yao, Haiyan Xie, George Mnatzaganian.

**Project administration:** Fan He, Irene Blackberry, Liqing Yao, George Mnatzaganian.

**Resources:** Irene Blackberry, Liqing Yao, George Mnatzaganian.

**Software:** Fan He.

**Supervision:** Irene Blackberry, Liqing Yao, George Mnatzaganian.

**Writing – original draft:** Fan He.

**Writing – review & editing:** Fan He, Irene Blackberry, Liqing Yao, Haiyan Xie, Tshepo Rasekaba, George Mnatzaganian.

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
