## [Decision Letter · Decision Letter 0]

27 Sep 2021

PONE-D-21-22996Pooled incidence and case-fatality of acute stroke in China: a systematic review and meta-analysisPLOS ONE

Dear Dr. He,

Thank you for submitting your manuscript to PLOS ONE. After careful consideration, we feel that it has merit but does not fully meet PLOS ONE’s publication criteria as it currently stands. Therefore, we invite you to submit a revised version of the manuscript that addresses the points raised during the review process.

 A few minor suggestions are already missing, find attached the reviewer comments (which could be suitable for the discussion section when justifying your findings); the manuscript improved in terms of the quality of the analysis, and most of the changes/suggestions were addressed.  Please submit your revised manuscript by Nov 11 2021 11:59PM. If you will need more time than this to complete your revisions, please reply to this message or contact the journal office at plosone@plos.org. Please include the following items when submitting your revised manuscript:A rebuttal letter that responds to each point raised by the academic editor and reviewer(s). You should upload this letter as a separate file labeled 'Response to Reviewers'.A marked-up copy of your manuscript that highlights changes made to the original version. You should upload this as a separate file labeled 'Revised Manuscript with Track Changes'.An unmarked version of your revised paper without tracked changes. You should upload this as a separate file labeled 'Manuscript'.If applicable, we recommend that you deposit your laboratory protocols in protocols.io to enhance the reproducibility of your results. Protocols.io assigns your protocol its own identifier (DOI) so that it can be cited independently in the future. For instructions see: https://journals.plos.org/plosone/s/submission-guidelines#loc-laboratory-protocols. Additionally, PLOS ONE offers an option for publishing peer-reviewed Lab Protocol articles, which describe protocols hosted on protocols.io. Read more information on sharing protocols at https://plos.org/protocols?utm_medium=editorial-email&utm_source=authorletters&utm_campaign=protocols.

We look forward to receiving your revised manuscript.

Kind regards,

Miguel A. Barboza, MD, MSc

Academic Editor

PLOS ONE

Journal Requirements:

Reviewers' comments:

Reviewer's Responses to Questions

**Comments to the Author**

1. Is the manuscript technically sound, and do the data support the conclusions?

Reviewer #1: Yes

2. Has the statistical analysis been performed appropriately and rigorously? 

Reviewer #1: Yes

3. Have the authors made all data underlying the findings in their manuscript fully available?

Reviewer #1: Yes

4. Is the manuscript presented in an intelligible fashion and written in standard English?

Reviewer #1: Yes

5. Review Comments to the Author

Reviewer #1: � The systematic review and meta-analysis is very well structured, ideas are clear and the writing is concise. I congratulate the author/authors for this. The review of literature is comprehensive and the importance of the research is very well discussed as far as applied perspective and theoretical aspect is concerned. The author/ authors have done good reference to the theoretical fundament for the article.

The tables, figures are helpful to understand the content of the paper.

I will like to suggest few points:

1. The article mentions that there is varying incidence of acute stroke in the the four economic regions with the lowest rates detected in the east coast. This could be because of better healthcare facilities and affordability of population in that region. Increased trends over time in the incidence of TS and IS were observed possibly because of major changes in lifestyle over a period of 4 decades, A three-to-twelve-month fatality following IS increased over time. This could be because of poor rehabilitation, affordability, inadequate follow up and non-compliance with the treatment. This could be because of poor counselling and information to the caregivers about the post stroke care of the patient.

2. This meta-analysis will help in:

a. Reinforcement and concretisation of policies by the policymakers in China to plan according to social and cultural variations of their four economic regions.

b. Planning strategies for primordial, primary and secondary prevention and allocating funds accordingly for infrastructural development.

c. Employing trained workforce in the field of neurosciences for stroke population. Also, developing multi-task force which will comprise of trained healthcare workers, community leaders, private and government sectors and other non-governmental organisations in China.

6. PLOS authors have the option to publish the peer review history of their article (what does this mean?). If published, this will include your full peer review and any attached files.

Reviewer #1: **Yes: **Dr.(Professor). Man Mohan Mehndiratta

---

## [Author Response · Author response to Decision Letter 0]

19 Oct 2021

20th October 2021

Dear Prof. Miguel A. Barboza,

Re: PONE-D-21-22996 – “Pooled incidence and case-fatality of acute stroke in China: a systematic review and meta-analysis”

Thank you for the opportunity to revise the manuscript which the reviewer described as “well structured” and “comprehensive” highlighting the importance of our research project as being “very well discussed”. In line with the editor’s and reviewer’s comments, as requested, we have responded on a point-by-point basis and have changed the manuscript accordingly.

Journal Requirements:

Response: The manuscript has been modified against these style requirements.

Response: As suggested, the reference list has been checked ensuring it is complete and correct. Retracted papers have not been cited in the manuscript.

The reference list has been revised to support the added points in the discussion as suggested by the reviewer. The reference of the accepted protocol of this study has also been added.

Response to reviewers’ comments

Reviewer #1: 

Comment 1:

The systematic review and meta-analysis is very well structured, ideas are clear and the writing is concise. I congratulate the author/authors for this. The review of literature is comprehensive and the importance of the research is very well discussed as far as applied perspective and theoretical aspect is concerned. The author/ authors have done good reference to the theoretical fundament for the article.

The tables, figures are helpful to understand the content of the paper.

Response: We thank the Reviewer for these positive comments.

Comment 2:

1. The article mentions that there is varying incidence of acute stroke in the the four economic regions with the lowest rates detected in the east coast. This could be because of better healthcare facilities and affordability of population in that region. Increased trends over time in the incidence of TS and IS were observed possibly because of major changes in lifestyle over a period of 4 decades. A three-to-twelve-month fatality following IS increased over time. This could be because of poor rehabilitation, affordability, inadequate follow up and non-compliance with the treatment. This could be because of poor counselling and information to the caregivers about the post stroke care of the patient.

Response: We thank the Reviewer for these insightful comments. Regional differences in stroke incidence in China have been continuously reported, often explained in the literature as a reflection of different prevalence of major risk factors, including hypertension, smoking, drinking and unhealthy diet. The unbalanced economic development in different regions may have impacted the health care system across different regions. Although populations residing in more developed regions (such as the East coast) may be utilizing health care services more than those residing in less developed regions. Following a comprehensive literature search we could not find any literature to support the notion that inequality in health resources distribution may have contributed to regional differences in stroke incidence. Therefore, we prefer to be more conservative when discussing these differences by only emphasizing the different patterns of risk factors across these regions. 

As suggested, the possible association between increasing stroke incidence and changing lifestyles is now discussed in the revised manuscript. As mentioned in the original manuscript, improving post-stroke rehabilitation may decrease the three-to-twelve-month fatality following ischaemic stroke (IS). In the revised manuscript we further discussed post stroke rehabilitation and its importance to study outcomes (please see page 19). Affordability is indeed a problem in stroke care in China, which may affect patients’ adherence to secondary stroke prevention medications and access to rehabilitation. The issue of affordability has also been added to this revised manuscript. Non-compliance reflected by nonadherence to medication and insufficient support for the caregivers have similarly been discussed. 

It is possible that inadequate follow up could have contributed to increasing three-to-twelve-month fatality following IS but we could not support this statement by published literature and therefore we did not add this point to the revised discussion. 

Comment 3

2. This meta-analysis will help in:

a. Reinforcement and concretisation of policies by the policymakers in China to plan according to social and cultural variations of their four economic regions.

b. Planning strategies for primordial, primary and secondary prevention and allocating funds accordingly for infrastructural development.

c. Employing trained workforce in the field of neurosciences for stroke population. Also, developing multi-task force which will comprise of trained healthcare workers, community leaders, private and government sectors and other non-governmental organisations in China.

Response: We thank the Reviewer for these insightful points, which we added to the revised discussion. 

Yours sincerely,

Fan He

La Trobe University, Melbourne

---

## [Decision Letter · Decision Letter 1]

25 Nov 2021

PONE-D-21-22996R1Pooled incidence and case-fatality of acute stroke in China: a systematic review and meta-analysisPLOS ONE

Dear Dr. He,

Thank you for submitting your manuscript to PLOS ONE. After careful consideration, we feel that it has merit but does not fully meet PLOS ONE’s publication criteria as it currently stands. Therefore, we invite you to submit a revised version of the manuscript that addresses the points raised during the review process.

 There are new suggestions regarding methodology and results section, according to our statistician expert, therefore, I suggest to pay attention to this new comments.

We look forward to receiving your revised manuscript.

Kind regards,

Miguel A. Barboza, MD, MSc

Academic Editor

PLOS ONE

Journal Requirements:

Reviewers' comments:

Reviewer's Responses to Questions

**Comments to the Author**

1. If the authors have adequately addressed your comments raised in a previous round of review and you feel that this manuscript is now acceptable for publication, you may indicate that here to bypass the “Comments to the Author” section, enter your conflict of interest statement in the “Confidential to Editor” section, and submit your "Accept" recommendation.

Reviewer #2: (No Response)

2. Is the manuscript technically sound, and do the data support the conclusions?

Reviewer #2: Partly

3. Has the statistical analysis been performed appropriately and rigorously? 

Reviewer #2: Yes

4. Have the authors made all data underlying the findings in their manuscript fully available?

Reviewer #2: Yes

5. Is the manuscript presented in an intelligible fashion and written in standard English?

Reviewer #2: No

6. Review Comments to the Author

Reviewer #2: I will focus on methods and reporting

Major

1) funnel plots and publication bias tests only make sense in the presence of an intervention, they are meaningless for incidence rates and in this context.

2) Language corrections are needed.

3) Some justification is needed on the 1990 time limit.

4) Meta-regression is a stab in the dark usually and is underpowered to detect anything but massive associations (effectively a regression with X observations, where X is the number of available studies). You should discuss this as a major limitation. Even with 60 or 80 studies, it can provide little insight.

5) Report the confidence intervals for I^2 (calculated using heterogi or metaan in Stata) as argued in http://www.ncbi.nlm.nih.gov/pubmed/17974687. A simple formula exists in the seminal 2002 Higgins paper that proposed I^2.

Minor

1) Add some more info on the methods section in the abstract, was heterogeneity assessed, publication bias assessed (if relevant) etc.

2) Clarify that the estimates were back transformed for the forest plots

3) The axis titles on the forest plots are not clear, reported as percentages when they are supposed to be rates. Currently the methods section and the graphs are not in agreement.

4) How was the random-effect model implemented, i.e. how was heterogeneity estimated? There are numerous ways to do so. Did they use the standard DerSimonian-Laird method? If so, please state so. Also there are better performing methods, for example please see https://www.ncbi.nlm.nih.gov/pubmed/28815652

5) What Stata commands were used? Again this relates to the point about the methods and the graphs.

6) Clarify the variables included in the meta regression in the methods section.

7. PLOS authors have the option to publish the peer review history of their article (what does this mean?). If published, this will include your full peer review and any attached files.

Reviewer #2: No

---

## [Author Response · Author response to Decision Letter 1]

24 Dec 2021

24th December 2021

Dear Prof. Miguel A. Barboza,

Re: PONE-D-21-22996R1 – “Pooled incidence and case-fatality of acute stroke in Mainland China, Hong Kong, and Macao: a systematic review and meta-analysis”

Thank you for the opportunity to revise the manuscript. In line with the editor’s and reviewer’s comments, as requested, we have responded on a point-by-point basis and have changed the manuscript accordingly.

Journal Requirements:

Response: As suggested, the reference list has been checked ensuring it is complete and correct. Retracted papers have not been cited in the manuscript.

We slightly revised the title of this manuscript making it consistent with the protocol paper that is currently in print. 

• He F, Blackberry I, Yao L, Xie H, Mnatzaganian G. Geographic Disparities in Pooled Stroke Incidence and Case Fatality in Mainland China, Hong Kong and Macao: Protocol for a Systematic Review and Meta-Analysis. JMIR Res Protoc. Forthcoming. doi:10.2196/32566

Response to reviewers’ comments

Comment 1:

If the authors have adequately addressed your comments raised in a previous round of review and you feel that this manuscript is now acceptable for publication, you may indicate that here to bypass the “Comments to the Author” section, enter your conflict of interest statement in the “Confidential to Editor” section, and submit your "Accept" recommendation.

Reviewer #2: (No Response)

No Response

Comment 2:

2. Is the manuscript technically sound, and do the data support the conclusions? The manuscript must describe a technically sound piece of scientific research with data that supports the conclusions. Experiments must have been conducted rigorously, with appropriate controls, replication, and sample sizes. The conclusions must be drawn appropriately based on the data presented. 

Reviewer #2: Partly

Response: We have responded to this comment. Please find our reasoning and responses detailed in the subsequent reviewer’s comments below 

Comment 3

Has the statistical analysis been performed appropriately and rigorously? 

Reviewer #2: Yes

Response: We thank the Reviewer for this positive comment.

Comment 4

Have the authors made all data underlying the findings in their manuscript fully available?

Reviewer #2: Yes

Response: We thank the Reviewer for this positive comment.

Comment 5

Is the manuscript presented in an intelligible fashion and written in standard English?

Reviewer #2: No

Response: This manuscript has been edited by a native English speaker. 

Comment 6

Reviewer #2: I will focus on methods and reporting

No Response

Major

1) funnel plots and publication bias tests only make sense in the presence of an intervention, they are meaningless for incidence rates and in this context.

Response: We respectfully disagree with the reviewer.

We used funnel plots together with Egger’s tests to investigate whether the studies included in the systematic review had a biased representation of higher or lower incidences; that is, publication bias. 

One criticism of meta-analyses is that often the available studies may not be representative of all studies addressing the research question. Publication bias refers to the claim that studies with statistically significant results are more likely to be published than studies without. Publication bias can occur in other instances besides trial study designs (Copper et al 2009). An example of a possible scenario is when a hospital director opposes the publication of results that show very high fatality rates. The funnel plot and Egger’s tests investigated whether our findings on between study heterogeneity was due to publication bias.

• Cooper H et al. The handbook of research synthesis and meta-analysis. 2nd edition. Russel Sage Foundation. New York. 2009

Running funnel plots with Egger’s test or other tests that investigate asymmetry is very commonly done in systematic reviews in observational incidence studies. We refer the reviewer to the following publications, which are a few of many that followed this very acceptable statistical method.

• Megan R et al. Systematic review and meta-analysis of the incidence rate of Takayasu arthritis. Rheumatology, 2020; 60(11):4982-4990.

• Makin SDJ, et al. Cognitive impairment after lacunar stroke: systematic review and meta-analysis of incidence, prevalence and comparison with other stroke subtypes. J Neurol Neurosurg Psychiatry. 2013; 84:893-900.

• Hackett ML, et al. Part I: frequency of depression after stroke: an updated systematic review and meta-analysis of observational studies. Int J Stroke. 2014, 9:1017-25.

• Li M et al. Hyperuricemia and risk of stroke: A systematic review and meta-analysis of prospective studies. Atherosclerosis, 2014:265-270.

• Taylor M et al. Longevity of complete dentures: a systematic review and meta-analysis. J Prosthetic Dentistry. 2021; 125:611-617

• Fasugba O et al. Ciprofloxacin resistance in community- and hospital acquired Escherichia coli urinary tract infections: a systematic review and meta-analysis of observational studies. BMC Infect Dis. 2015, 15:545.

• Qin B, et al. Epidemiology of primary Sjögren's syndrome: a systematic review and meta-analysis. Annals of the Rheumatic Diseases 2015;74:1983-1989.

• Meng K, et al. "Incidence of venous thromboembolism during pregnancy and the puerperium: a systematic review and meta-analysis." The Journal of Maternal-Fetal & Neonatal Medicine 28.3 (2015): 245-253.

2) Language corrections are needed.

Response: This manuscript has been edited by a native English speaker.

3) Some justification is needed on the 1990 time limit.

Response: The decision to exclude studies published before 1991 was made to better capture studies that followed similar diagnostic criteria. Before 1991, clinicians would have diagnosed stroke without brain imaging technologies which could have resulted in the inclusion of false positive cases. The restriction to 1991 onwards aimed to decrease false positives and to increase the validity of included cases. We have added a clarification in the revised manuscript.

4) Meta-regression is a stab in the dark usually and is underpowered to detect anything but massive associations (effectively a regression with X observations, where X is the number of available studies). You should discuss this as a major limitation. Even with 60 or 80 studies, it can provide little insight.

Response: Meta regression offers a systematic approach to explain sources of wide variation in reported findings. In our incidence studies, no heterogeneity was detected and, therefore, there was no need to conduct further analyses. However, in the case-fatality analyses, we conducted meta regression to explain between study variances based on the available study variables.

Meta-regression is an extension to subgroup analyses that allows the effect of continuous, as well as categorical characteristics to be investigated, and in principle allows the effects of multiple factors to be investigated simultaneously (although this is rarely possible due to inadequate numbers of studies) (Thompson 2002). According to scientific literature, meta-regression should generally not be considered when there are fewer than 10 to 15 studies in a meta-analysis. Our case-fatality analysis included 39 studies.

1. Cochrane handbook for systematic reviews of interventions. https://training.cochrane.org/handbook/current

2. Higgins JPT, et al. Measuring inconsistency in meta-analysis. BMJ Clinical Research. 327:557-60.

3. Thompson SG, et al. How should meta-regression analyses be undertaken and interpreted? Statistics in Medicine. 2002; 21:1559-1573

4. https://handbook-5-1.cochrane.org/chapter_9/9_6_4_meta_regression.htm

We acknowledge that exploring all sources of heterogeneity is limited as there are many unknown variables that cannot be accounted for. This is always true in all systematic reviews (Copper et al, 2009). Furthermore, the limited number of included studies in the meta-regression add uncertainty in the true sources of heterogeneity. These limitations have been added to the revised manuscript.

5) Report the confidence intervals for I^2 (calculated using heterogi or metaan in Stata) as argued in http://www.ncbi.nlm.nih.gov/pubmed/17974687. A simple formula exists in the seminal 2002 Higgins paper that proposed I^2.

Response: The heterogeneity in case-fatality analyses was very high and statistically significant (p<0.001). As requested, using Borenstein and colleagues methods, we have estimated confidence intervals for the statistic I-squared..

Borenstein M, et al. Introduction to meta-analysis. John Wiley & Sons, Ltd. UK. 2011

Minor 

1) Add some more info on the methods section in the abstract, was heterogeneity assessed, publication bias assessed (if relevant) etc.

Response: As suggested, we have revised the abstract to include this. 

2) Clarify that the estimates were back transformed for the forest plots

Response: The methods we used are stated in the manuscript: “Using the Exact method, the incidence rates of disease were expressed as Poisson means divided by time at risk. The log incidence rates together with their corresponding log standard errors, stratified by the four economic regions, were meta-analysed using DerSimonian and Laird random effects model. Random effects models were also used to estimate the pooled case fatality estimates”.

3) The axis titles on the forest plots are not clear, reported as percentages when they are supposed to be rates. Currently the methods section and the graphs are not in agreement.

Response: As shown in Figure 2, the title of axis is incidence rate. While, in Figures 4 and 5, the title of axis is proportion. 

4) How was the random-effect model implemented, i.e. how was heterogeneity estimated? There are numerous ways to do so. Did they use the standard DerSimonian-Laird method? If so, please state so. Also there are better performing methods, for example please see https://www.ncbi.nlm.nih.gov/pubmed/28815652

Response: This has been stated in the section of “Statistical analysis”: “The log incidence rates together with their corresponding log standard errors, stratified by the four economic regions, were meta-analysed using DerSimonian and Laird random effects model. Random effects models were also used to estimate the pooled case fatality estimates.” We have revised the forest plots showing the case fatalities – now including the weights that were generated by the random effect model. 

5) What Stata commands were used? Again this relates to the point about the methods and the graphs.

Response: We used metan and metaprop Stata commands for incidence and case fatality studies, respectively. We have added this clarification in Methods.

6) Clarify the variables included in the meta regression in the methods section.

Response: The meta-regression models included risk of bias, study setting (community or hospital), follow-up time, smoking status, age, sex, and severity of stroke. This has been clarified in the methods section. 

Yours sincerely,

Fan He

La Trobe University, Melbourne

---

## [Editor Report · Decision Letter 2]

14 Jun 2022

Pooled incidence and case-fatality of acute stroke in Mainland China, Hong Kong, and Macao: a systematic review and meta-analysis

PONE-D-21-22996R2

Dear Dr. He,

We’re pleased to inform you that your manuscript has been judged scientifically suitable for publication and will be formally accepted for publication once it meets all outstanding technical requirements.

Kind regards,

Salvatore De Rosa

Academic Editor

PLOS ONE

---

## [Editor Report · Acceptance letter]

17 Jun 2022

PONE-D-21-22996R2 

Pooled incidence and case-fatality of acute stroke in Mainland China, Hong Kong, and Macao: a systematic review and meta-analysis 

Dear Dr. He:

I'm pleased to inform you that your manuscript has been deemed suitable for publication in PLOS ONE. Congratulations! Your manuscript is now with our production department. 

Kind regards, 

on behalf of

Dr. Salvatore De Rosa 

Academic Editor

PLOS ONE